# Homologous Recombination Repair Gene Alterations Are Associated with Tumor Mutational Burden and Survival of Immunotherapy

**DOI:** 10.3390/cancers15235608

**Published:** 2023-11-27

**Authors:** Mamoru Ito, Makoto Kubo, Hitomi Kawaji, Yoshiki Otsubo, Kanako Kurata, Hikaru Abutani, Mikita Suyama, Yoshinao Oda, Tomoharu Yoshizumi, Masafumi Nakamura, Eishi Baba

**Affiliations:** 1Department of Hematology, Oncology and Cardiovascular Medicine, Kyushu University Hospital, Fukuoka 812-8582, Japan; ito.mamoru.632@m.kyushu-u.ac.jp; 2Department of Breast Surgical Oncology, Kyushu University Hospital, Fukuoka 812-8582, Japan; 3Department of Surgery and Oncology, Graduate School of Medical Sciences, Kyushu University, Fukuoka 812-8582, Japan; 4Foundation Medicine Unit, Foundation Medicine Business Department, Chugai Pharmaceutical Co., Ltd., Tokyo 103-8324, Japan; 5Division of Bioinformatics, Medical Institute of Bioregulation, Kyushu University, Fukuoka 812-8582, Japan; 6Department of Anatomic Pathology, Graduate School of Medical Sciences, Kyushu University, Fukuoka 812-8582, Japan; 7Department of Surgery and Science, Graduate School of Medical Sciences, Kyushu University, Fukuoka 812-8582, Japan; 8Department of Oncology and Social Medicine, Faculty of Medical Sciences, Kyushu University, Fukuoka 812-8582, Japan

**Keywords:** homologous recombination repair gene, *ARID1A*, tumor mutational burden, comprehensive genomic profiling

## Abstract

**Simple Summary:**

A multigene panel test, known as comprehensive genomic profiling (CGP), is now available in Japan for advanced cancer patients to identify suitable genotype-matched therapies. However, well-established biomarkers to guide these genotype-matched therapies are limited. Test failures can arise due to clinical factors affecting DNA yield and quality. Consequently, only 8% of patients undergoing CGP can access matched therapies. To address this issue, a retrospective analysis of clinical data from patients who underwent the FoundationOne^®^ CDx at Kyushu University Hospital was conducted. We found that alterations in the homologous recombination repair (HRR) genes were associated with a high mutational burden. Data from public sources suggested that patients with HRR gene alterations had higher TMB and showed significantly longer survival of immunotherapy. Although immunotherapy has a key role in cancer treatment, predicting its efficacy through biomarkers remains challenging. Investigating HRR gene alterations could help select patients who are more likely to benefit from immunotherapy.

**Abstract:**

Background: Comprehensive genomic profiling (CGP) has become generally accepted practice in cancer care since CGP has become reimbursed by national healthcare insurance in Japan in 2019. However, its usefulness for cancer patients is insufficient for several reasons. Methods: In an observational clinical study of FoundationOne^®^ CDx, potential biomarkers were explored and the cause of testing failure was investigated. A total of 220 cancer patients were enrolled in the study during the period from 2018 to 2019 at Kyushu University Hospital. Results: The primary tumor sites of the 220 cases were breast (115), colon (29), stomach (19), and pancreas (20). The present dataset suggested that homologous recombination repair (HRR) gene alterations were positively associated with tumor mutational burden-high (TMB-high) (*p* = 0.0099). A public dataset confirmed that patients with HRR gene alterations had a higher TMB and showed significantly longer survival of immunotherapy. In the present study, 18 cases failed sequencing. A lower percentage of tumor cell nuclei was the most common reason for testing failures (*p* = 0.037). Cases that received neoadjuvant chemotherapy before sampling tended to fail testing. Conclusions: HRR gene alterations can be a potential biomarker predicting TMB-high and a good response to immunotherapy. For successful sequencing, samples with lower percentages of tumor cell nuclei and previous neoadjuvant chemotherapy should be avoided.

## 1. Introduction

Cancer is the leading cause of death worldwide. Although the development of anticancer drugs for advanced cancer has been remarkable, the life expectancy of patients with advanced cancer is still only a few years. However, the recent progress of molecular techniques enabled multi-omics analysis to enhance our understanding cancer biology. In particular, next-generation sequencing (NGS) technology has revolutionized the simultaneous assessment of hundreds or even thousands of genes, not only in clinical research but also in clinical care [1,2]. Recently, it was reported that treatment with molecular targeted therapies based on comprehensive genomic profiling (CGP) significantly improved survival outcomes for patients with pancreatic cancer compared with patients who received non-molecularly matched therapies [3]. Given this result, the National Comprehensive Cancer Network (NCCN) guideline stated that not only genetic testing for inherited mutations, but also CGP via FDA-approved and/or validated next-generation sequencing (NGS) based assays were recommended for patients with locally advanced/metastatic pancreatic cancer [4]. Although there are not many molecular biomarkers that lead to effective treatment, the introduction of precision oncology can have a substantial effect on survival in patients with various solid tumors.

CGP tests, which promote genotype-matched therapy for advanced cancer, were covered by national healthcare insurance by The Ministry of Health, Labor, and Welfare (MHLW) in 2019, and they are available in daily routine clinical care in Japan. Currently, three types of CGP tests, two tissue-based tests, FoundationOne^®^ CDx and NCC Oncopanel [5], and one blood-based test, FoundationOne^®^ Liquid CDx, are available. These tests have two functions. One is as companion diagnostics for the use of certain drugs, and the other is for CGP. However, unfortunately, the CGP test can currently only be used once in a patient’s lifetime due to restrictions on reimbursement. Therefore, it is important to avoid test failure because several clinical factors can easily lead to sequencing failure.

Even when the tests are successfully conducted, the ratio of patients who can proceed to matched therapies based on CGP is still low. The Center for Cancer Genomics and Advanced Therapeutics (C-CAT) reported only 8.1% of patients received matched therapies via CGP. To maximize the utility of CGP, it is essential not only to develop new drugs, but also to establish new biomarkers. The development of cancer immunotherapy, represented by PD-1/PD-L1 blockade, is emerging regardless of cancer type. Currently, patients with most cancer types can receive cancer immunotherapy, but the objective response rate depends on the cancer type. While PD-L1 expression is the most common biomarker for predicting responses to PD-1/PD-L1 blockade therapy, it alone is insufficient for selecting patients who benefit from immunotherapy [6,7]. Patients with microsatellite instability-high (MSI-high) and/or tumor mutational burden-high (TMB-high) are expected to have good responses, but biomarkers of cancer immunotherapy are still not well known [8,9,10,11,12,13].

Homologous recombination (HR) is one of the essential processes for the repair of DNA double-strand breaks. *BRCA1* and *BRCA2* genes are representative genes involved in the HR process. There are several key components for HR other than *BRCA1/2*, which are the so-called HR-repair genes (HRR genes), and some HRR gene alterations in germlines cause hereditary cancer syndromes. Recent clinical trials showed that the loss of function of HRR genes in cancer (homologous recombination deficiency: HRD) can be a therapeutic target of poly (ADP-ribose) polymerase (PARP) inhibitors by synthetic lethality [14,15]. *ARID1A* is one of the SWI/SNF chromatin remodeling complex subunits. *ARID1A* acts as a tumor suppressor and is frequently mutated in various cancer types. Recent research showed that *ARID1A* has important roles in the DNA damage response. ARID1A-containing SWI/SNF complexes allow DNA repair proteins to access DNA damage sites and help repair double-strand breaks (DSB) via non-homologous end joining (NHEJ) and HR [16,17]. Furthermore, Shen and colleagues reported that *ARID1A* also interacts with *MSH2*, which is one of the mismatch repair genes (MMR genes), and recruits MSH2 to chromatin during DNA replication. *ARID1A* deficiency compromises MMR and leads to increased mutagenesis [18]. It is not well known whether HRR gene alterations can be the target of anti-cancer drugs other than PARP inhibitors.

In the present study, the results of CGP conducted as clinical research in our institute were reviewed. First, the aim was to develop a precision oncology system to complete CGP and benefit patients with unresectable/metastatic disease with the CGP results. The molecular biomarkers that predict the effectiveness of cancer immunotherapy were also explored, and HRR gene alterations can be proposed as biomarkers of cancer immunotherapy (UMIN000050577).

## 2. Materials and Methods

### 2.1. Patient Population

A total of 220 solid tumor patients treated at Kyushu University Hospital were enrolled in this study. All patients provided their written, informed consent for participation. The study conformed to the principles of the Declaration of Helsinki and was approved by the Institutional Review Board of Kyushu University Hospital (No. 758-00 and 768-00).

### 2.2. Cancer Genomic Testing with an NGS-Based Multiplex Gene Panel Assay (FoundationOne^®^ CDx)

FoundationOne^®^ CDx (Foundation Medicine Inc., Cambridge, MA, USA) was used as a targeted multiplex cancer panel test for research purposes only. All the tests in the present study were conducted before MHLW approval of FoundationOne^®^ CDx in Japan. FoundationOne^®^ CDx is an NGS-based comprehensive genomic profiling tool for the detection of nucleotide substitutions (SUBs), insertion and deletion alterations (indels), and copy number alterations (CNAs) in 324 genes and selected gene rearrangements, as well as genomic signatures including microsatellite instability (MSI) and tumor mutational burden (TMB) (Appendix A).

DNA samples for testing were extracted from formalin-fixed, paraffin-embedded tumor tissue specimens. Samples were prepared according to the manufacturer’s instructions. Usually, 10 unstained slides, with a thickness of 4–5 µm and total tumor volume of more than 1 mm^3^, were required. The optimal percentage of tumor nuclei was at least 30%, and a minimum of 20% was required. The clinical physician chose the specimen for testing, and then pathologists assessed sample suitability and prepared the slides.

### 2.3. Statistical Analysis and Generating Figures

Statistical analysis was performed with JMP Pro 14 (SAS Institute Inc., Cary, NC, USA). The oncoprint in Figure 1 was generated by Oncoprinter in cBioportal [19,20]. The open genetic dataset of advanced cancer patients who received immunotherapy was also quoted from cBioportal and was originally reported by Samstein et al., from Memorial Sloan Kettering Cancer Center (MSK IMPACT dataset) [21].

## 3. Results

### 3.1. Patients’ Characteristics

Patients’ background characteristics are summarized in Table 1. A total of 220 cancer patients were enrolled, of whom 204 were successfully tested by FoundationOne^®^ CDx, 18 cases failed the tests, and two resubmitted another specimen after initial failure. The median age of the participants was 63 years. The tests were performed in the advanced stage in 128 cases and after complete resection in 92 cases. Cancer types were breast in 109 cases, colorectal in 26 cases, and gastric in 19 cases. The submitted specimens were collected from the primary site in 158 cases and from metastatic sites in 46 cases.

### 3.2. Genomic Characterization

According to the profiling results of all participants, *TP53* was the most common alteration (58%). Alterations in *PIK3CA* (28%), *KRAS* (20%), and *APC* (13%) followed (Figure 1). The genomic landscape of breast cancer has been previously described [22]. In colorectal cancer patients, genomic alterations in *TP53*, *APC*, *KRAS*, and *SMAD4* were frequently observed. Alterations in *TP53*, *KRAS*, *ARID1A*, and *APC* in gastric cancer patients and alterations in *KRAS*, *TP53*, *SMAD4*, *CDKN2A*, and *CDKN2B* in pancreatic cancer patients were observed (Table 2). Seven soft tissue sarcoma cases were submitted, and dedifferentiated liposarcoma was the most frequent pathological diagnosis; therefore, *CDK4* and *MDM2* amplification were frequently observed among them.

### 3.3. Cases with MMR Deficient (dMMR), HRR Mutant (HRRm), MSI-High, and TMB-High

MSI-high was detected in only 4 cases (2.0%). Twenty-one cases (10.2%) harbored 10 or more mutations/Mb in coding regions, which were defined as tumor mutational burden-high (TMB-high) cases (Figure 2). In MSI-high cases, mismatch repair gene (*MLH1*, *MSH2*, *MSH6*, and *PMS2*) alterations was detected only in one case (*MSH2* mutant), and this case was diagnosed with Lynch syndrome by subsequent germline testing (Appendix A). Alterations of MMR genes and HRR genes in TMB-high cases were extracted (Figure 3A), TMB-high seemed to be associated not only with MMR gene alterations, but also with HRR gene alterations (Figure 3B, Table 3).

To confirm that HRR gene alterations are associated with TMB-high, another large data set from the cBioportal dataset was examined [19,20,21]. The MSK-IMPACT dataset comprises 1662 patients who received immunotherapy. Among them, 1256 received anti-PD-1 or PD-L1, 146 received anti-CTLA-4, and 260 received a combination of anti-CTLA-4 and anti-PD-1/PD-L1 therapies. This dataset from MSK-IMPACT suggests that a considerable proportion of patients with TMB-high possessed HRR gene alterations (Figure 3C). Most patients with TMB-high in our dataset did not receive immunotherapy due to their cancer types; therefore, the association between HRR gene alterations and survival on immunotherapy was examined in the MSK-IMPACT dataset. Patients with HRR-mutants (HRRm) had higher TMB and showed significantly longer survival (Figure 3D,E). In the present study, *ARID1A* mutant cases tend to show TMB-high. Focusing on *ARID1A* among the HRR genes, *ARID1A* mutants also had higher TMB and showed longer survival in the MSK-IMPACT dataset (Figure 3F,G).

### 3.4. Clinical Factors Affecting Testing Failure

Next, clinical factors affecting sequencing failure were investigated. Eighteen cases in the present dataset failed testing. The sampling method and tumor site were not associated with testing success or failure (resection vs. biopsy, *p* = 0.60; primary vs. metastasis, *p* = 0.76). Cases on neoadjuvant chemotherapy before sampling tended to fail sequencing, but not significantly (*p* = 0.22) (Table 4). A low percentage of tumor cell nuclei caused significant sequencing failure (*p* = 0.037, Figure 4).

### 3.5. Presumed Germline Pathogenic Variants (PGPVs) Assessment and the Low Rate of Germline Testing in Patients with PGPVs

Although the primary objective of CGP is the identification of biomarkers to guide genotype-matched therapy, tumor genomic testing may also reveal germline gene alterations that are linked to heritable cancer susceptibility and other conditions. These potentially heritable gene alterations found in tumor-only sequencing tests are called “presumed germline pathogenic variants (PGPVs)”, and their origin should be confirmed thorough subsequent germline testing [23]. The reporting of PGPVs and the referral to genetic counseling is one of the important tasks of the molecular tumor board, called the “expert panel” in Japan. Our institute originally defined a gene list of recommendations for the reporting of PGPVs. These gene lists were based on ACMG and ESMO recommendations [24]. Genes with a heritable predisposition to cancer were divided into two categories, Group A and Group B (Table 5). We recommend genetic counseling for patients with group A genes alterations regardless of family history or their own history of cancer, because it is highly suspected that group A gene alterations derive from germline alterations. According to ESMO recommendations, these gene alterations in tumors indicate that they originated from germline alterations with over 10% probability, that is, group A gene alterations have a high germline conversion rate. On the other hand, group B gene alterations can originate from germline alterations, but the probability is relatively low. When patients harboring group B gene alterations in tumors have a certain personal history or family history, the germline conversion rate can exceed 10%. Therefore, we decide whether to recommend genetic counseling for patients with group B gene alterations considering their history. In the present study, 43 cases had Group A gene alterations, and genetic counseling was recommended to them. *BRCA1* alteration was the most frequent gene alteration among them (Figure 5). However, only four cases had genetic counseling. The others did not have genetic counseling despite physicians’ recommendations.

## 4. Discussion

CGP is becoming essential in medical oncology for diagnosis, classification, and choice of treatment, serving as an important tool for clinical decision support. Although the national health insurance system in Japan reimbursed CGP in clinical care, CGP is available in limited dedicated hospitals and not all patients are tested at this time. The MHLW designated 12 Core Hospitals for Cancer Genomic Medicine and more than 100 Cooperative Hospitals for Cancer Genomic Medicine in September 2022. Now is a transitional period for CGP to become common in Japan. We should maximize the usefulness of CGP for patients without effective choices of treatment.

FoudationOne^®^ CDx is one of the NGS-based CGP tests utilized globally, and its clinical validation data is extensively documented elsewhere [25]. This test is exclusively performed by Foundation Medicine as a central laboratory testing service, adhering to the regulations of the College of American Pathologists (CAP) and Clinical Laboratory Improvement Amendments (CLIA). The concordance between FoundationOne^®^ CDx and the validated orthogonal comparator assays has been assessed, and the positive percent agreement (PPA) is reported to range from 89.4% to 100%, depending on biomarkers. The limit of detection (LoD) was determined based on either allele frequency or tumor purity. The LoD allele fraction for SUBs and indels was in the range of 2.0% to 12.74%, and the LoD tumor purity for copy number alterations and genomic rearrangements was in the range of 1.8% to 30%. The reproducibility of calling genomic alterations exceeded 99%. In the present study, the FoudationOne^®^ CDx results of 220 cancer patients were retrospectively reviewed, and an attempt was made to identify a new predictive biomarker for medical treatment. Problems in obtaining adequate test results were also investigated.

The present study showed the association between HRR gene alterations and TMB-high. As described in the introduction, HRR gene alterations can cause mutagenesis in several ways. *CDK12* is one of the representative HRR genes, and Wu and colleagues reported that *CDK12*-mutant prostate cancer produces neo-fusion genes, and these neo-fusion genes can be neo-antigens for the immune system [26]. A small cohort of *CDK12*-mutant prostate cancer cases suggested a minor but positive effect of PD-1 blockade therapy [27]. However, neo-fusion cannot be detected unless RNA sequencing is performed. Therefore, neo-fusion assessment is not available because current commercial CGP tests in Japan are basically based on DNA sequencing technologies. Although there are qualitative differences between single nucleotide variants and neo-fusion, the present study supports HRR gene-mutant cancer also showing TMB-high, and HRR mutants can be the target of PD-1/PD-L1 blockade therapy. Of HRR genes, *ARID1A* was found to be frequently mutated in the present dataset and the MSK-IMPACT dataset. Consistent with the present results, some studies reported that HRR genes or *ARID1A* alteration can be a biomarker that predicts sensitivity to PD-1/PD-L1 blockade therapy [28,29].

Whether HRR gene alterations are a true predictive biomarker for immunotherapy can depend on the cancer type. For example, Hugo reported that PD-1 blockade was more effective for *BRCA2*-mutant melanoma [30], but avelumab did not show a clinically meaningful effect for advanced ovarian carcinoma (JAVELIN Ovarian 100 trial), despite the fact that HRD is closely involved in the development of ovarian cancer [31]. Although subgroup analysis by genotype has not yet been conducted in the JAVELIN Ovarian 100 study, further analysis could provide insight into whether HRR mutants are a true target of PD-1/PD-L1 blockade therapy.

Currently, PARP inhibition is one of the established treatment options for BRCA-mutant cancers, but these cancers can overcome PARP inhibitors via “BRCA reversion”. BRCA reversion restores BRCA function by secondary mutations that skip the stop codon or deleterious mutations [32,33]. Adding PD-1/PD-L1 blockade therapy to PARP inhibitor therapy can regulate tumor activity even after the tumor overcomes PARP inhibition. Combination therapies of a PARP inhibitor and a PD-1/PD-L1 blockade are currently being evaluated in several clinical studies, and some of them are reporting hopeful results [34,35].

The present study also suggests several problems to solve in the CGP process, including sample quality and the management of PGPVs. Generally, the performance of diagnostic tests can be influenced by the method, region, and time period [36]. Although FoudationOne^®^ CDx centralizes testing to maintain quality, it’s important to acknowledge that pre-analytical factors can impact testing success [37]. In our practice, tumor content affected testing success or failure the most. In addition, pre-operative chemotherapy tended to cause testing failure, though not significantly. Currently, liquid biopsy, represented by FoundationOne^®^ Liquid CDx, is also available in clinical care. However, there are some pitfalls in liquid biopsy. Liquid biopsy is less sensitive in detecting genomic alterations, especially when the amount of circulating tumor DNA (ctDNA) is low. The amount of ctDNA depends on tumor load, tumor type, and metastasis site [38,39,40,41]. Tissue-based testing is still the standard method for CGP rather than liquid-based tests. We should try to obtain sufficient biopsy samples before conducting pre-operative chemotherapy.

Identifying pathogenic germline variants (PGVs) can inform future cancer risks, surveillance, and prevention options for patients and their family members. From this point of view, the Core Hospitals for Cancer Genomic Medicine in Japan are mandated to establish a genetic counseling system to enhance public health. The present study found 43 cases harboring PGPVs, and genetic counseling was recommended to the attending physicians and patients. However, only four patients had genetic counseling despite these promotions. The National Health Interview Survey conducted in the United States has revealed that both the region and time period significantly impacted the rate of genetic testing among individuals with risk factors for Lynch syndrome [42]. The low rate of genetic counseling in the present study can be attributed to limited awareness and a lack of attitudes toward genetic testing among patients [43]. Enhancing understanding of genetic counseling through education for patients and healthcare providers is imperative. It has gradually become common to receive genetic counseling and testing during cancer treatment in Japan, in contrast to the period of the present study, as treatment with PARP inhibitors has gained popularity. Additionally, the costs for identifying PGVs may serve as one of the reasons discouraging individuals from genetic counseling. The national health insurance system in Japan allows cancer patients to receive treatment without a high financial burden, but this public insurance system does not cover diagnosis and surveillance for most hereditary cancer syndromes except for hereditary breast and ovarian cancer syndrome. Therefore, patients with these syndromes must cover all costs for the care of hereditary cancer syndromes in most cases. This national health insurance system is still insufficient for early cancer detection in hereditary cancer syndrome patients in Japan.

The present study has highlighted various problems and limitations associated with CGP in Japan. However, the ongoing advancements in technology are expected to move precision medicine forward. Current CGP in clinical care is restricted to DNA analysis for select genes and is not able to reveal multi-omics processes involved in cancers, such as methylation, cytokines, and protein expression. Several recent clinical trials have sought to enhance outcomes for cancer patients by combining DNA, RNA, and DNA methylation analyses [1,44,45]. In addition, the analysis of cytokines and immune cells is becoming crucial for predicting the efficacy of immunotherapy [2,46,47]. The future of precision medicine demands a shift towards multi-omics approaches that go beyond DNA sequencing, that will bring a more comprehensive understanding of the biological processes underlying cancer and offer improved outcomes for patients.

## 5. Conclusions

HRR genes appear to be molecular biomarkers for cancer immunotherapy. However, it is important to complete and use CGP effectively and to avoid several clinical factors leading to testing failure, which are a lower percentage of tumor cell nuclei and previous neoadjuvant chemotherapy.

## Figures and Tables

**Figure 1 cancers-15-05608-f001:**
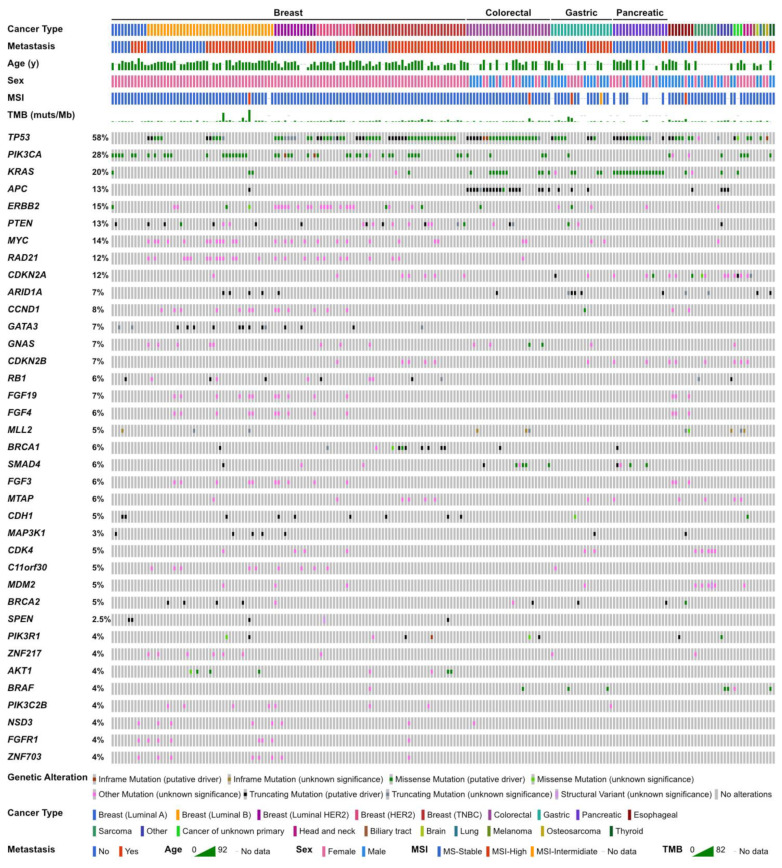
Genomic landscape of the present study.

**Figure 2 cancers-15-05608-f002:**
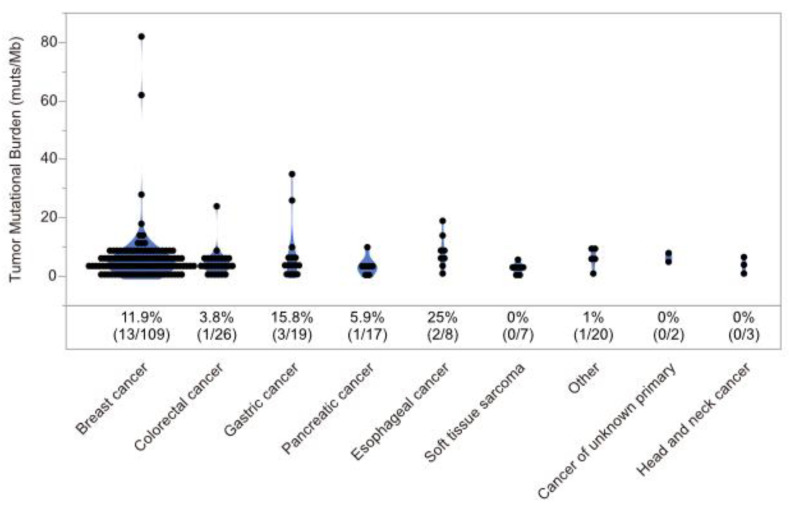
Distribution of tumor mutational burden. Percentages represent the ratio of cases with tumor mutational burden greater than 10 muts/Mb in coding regions.

**Figure 3 cancers-15-05608-f003:**
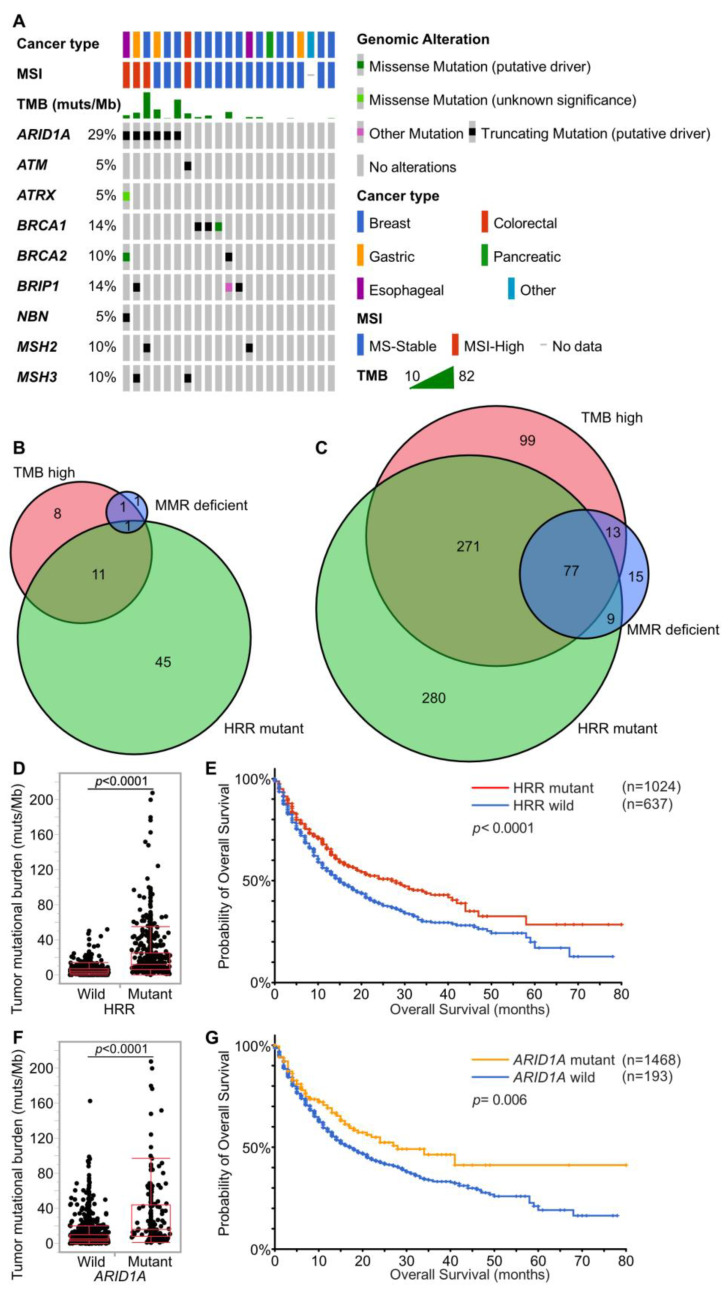
Associations of TMB, dMMR, and HRRm. (**A**). Associations of TMB, MSI, dMMR, and HRRm in the present study. The gene list of MMR genes and HRR genes is shown in Appendix A. (**B**,**C**) Venn diagram showing associations of cases with TMB-high (10 muts/Mb or more), MMR gene mutants, and HRR gene mutants. (**B**) is for the present study. (**C**) is for the MSK-IMPACT dataset [21]. (**D**,**E**) Distribution of TMB and survival on immunotherapy by HRR gene mutation in the MSK-IMPACT dataset. HRR mutant cases show significantly higher TMB and prolonged survival on immunotherapy (Student’s *t*-test and the log-rank test). (**F**,**G**) Distribution of TMB and survival on immunotherapy by *ARID1A* gene alteration in the MSK-IMPACT dataset. *ARID1A* mutant cases show significantly higher TMB and prolonged survival on immunotherapy (Student’s *t*-test and the log-rank test).

**Figure 4 cancers-15-05608-f004:**
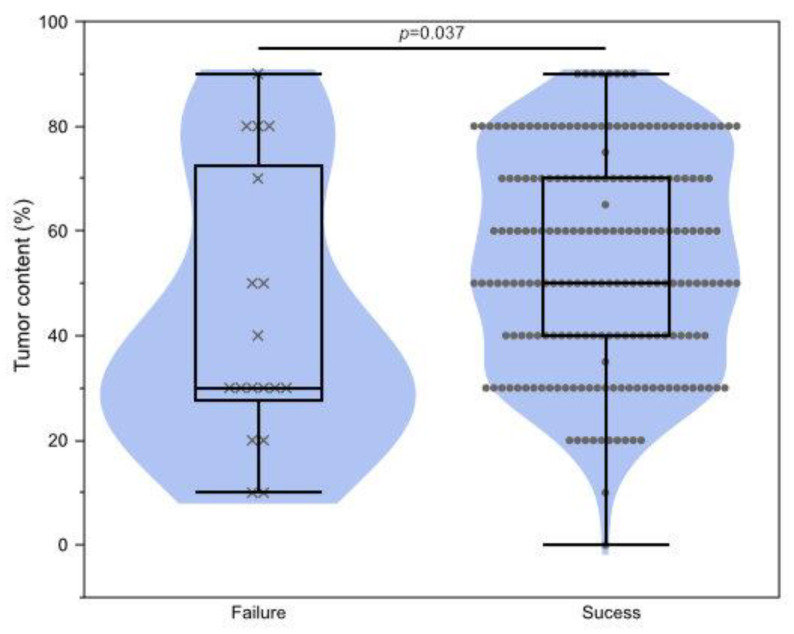
Association of testing failure and tumor content. Wilcoxon rank-sum test *p* = 0.037.

**Figure 5 cancers-15-05608-f005:**
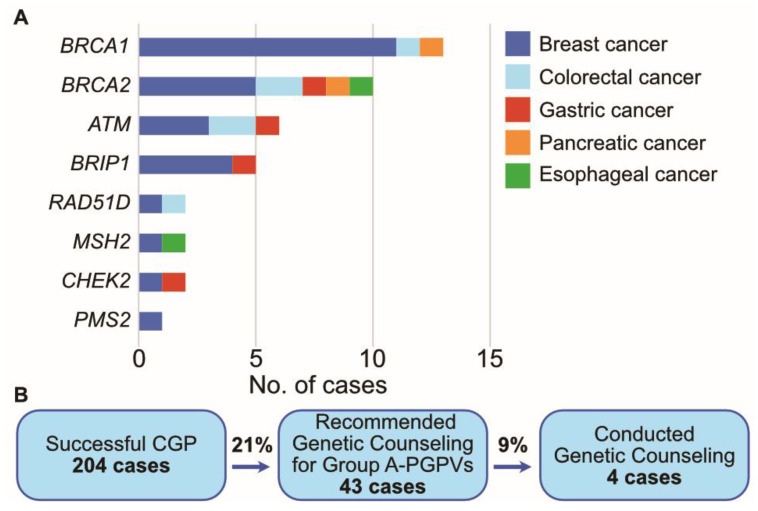
PGPV assessment in the present study. (**A**). The number of cases with PGPVs categorized in Group A. (**B**). Flow diagram of cases with PGPVs.

**Table 1 cancers-15-05608-t001:** Patients’ characteristics with successful tests (*n* = 204).

	Number of Patients (%)
Age, years, median (range)	63 (22–92)
Sex	
Male	54 (26%)
Female	150 (74%)
Disease	
Breast cancer	109 (53%)
Colorectal cancer	26 (13%)
Gastric cancer	19 (9%)
Pancreatic cancer	17 (8%)
Esophageal cancer	8 (4%)
Soft tissue sarcoma	7 (3%)
Other	5 (2%)
Head and neck cancer	3 (1%)
Cancer of unknown primary	3 (1%)
Thyroid cancer	2 (1%)
Lung cancer	1 (0.5%)
Osteosarcoma	1 (0.5%)
Brain cancer	1 (0.5%)
Biliary tract cancer	1 (0.5%)
Melanoma	1 (0.5%)
TNM Staging	
I	3 (1%)
II	42 (36%)
III	42 (36%)
IV	117 (57%)
Sampling method	
Biopsy	67 (33%)
Resection	137 (67%)
Sampling site	
Primary	158 (77%)
Metastatic	46 (23%)
Prior chemotherapy	
Yes	40 (20%)

**Table 2 cancers-15-05608-t002:** Top-ranked altered genes in each cancer type.

Breast (*n* = 109)		Colorectal (*n* = 26)		Gastric (*n* = 19)		Pancreatic (*n* = 17)	
*TP53*	61 (56.0%)	*TP53*	23 (88.5%)	*TP53*	8 (42.1%)	*KRAS*	16 (94.1%)
*PIK3CA*	43 (39.4%)	*APC*	19 (73.1%)	*KRAS*	5 (26.3%)	*TP53*	13 (76.5%)
*ERBB2*	25 (22.9%)	*KRAS*	10 (38.5%)	*ARID1A*	4 (21.1%)	*SMAD4*	4 (23.5%)
*MYC*	23 (21.1%)	*SMAD4*	6 (23.1%)	*APC*	3 (15.8%)	*CDKN2A*	3 (17.6%)
*RAD21*	23 (21.1%)	*GNAS*	4 (15.4%)	*ERBB2*	3 (15.8%)	*CDKN2B*	2 (11.8%)
*PTEN*	20 (18.3%)	*PIK3CA*	4 (15.4%)			*MET*	2 (11.8%)
*GATA3*	15 (13.8%)	*MLL2*	3 (11.5%)				

**Table 3 cancers-15-05608-t003:** Associations of TMB, MSI, dMMR, and HRRm in the present study.

	TMB-High *	TMB-Low	*p*-Value **
MSI			
MSI-high	4	0	<0.0001
MS-stable	17	166	
MMR			
MMR mutant	2	1	0.0337
MMR wild type	19	165	
HRR			
HRR mutant	12	45	0.0099
HRR wild type	9	121	

* TMB-high: TMB greater than 10 muts/Mb in coding regions ** Fisher’s exact test.

**Table 4 cancers-15-05608-t004:** Associations between clinical factors and sequencing failure in the present study.

	Failure (*n* = 18)	Success (*n* = 204)	*p*-Value *
Sampling method			0.60
Biopsy	7	67	
Resection	11	137	
Sampling site			0.76
Primary site	15	158	
Metastatic site	3	46	
Prior chemotherapy			0.22
Yes	6	40	
No	12	164	

* Fisher exact test.

**Table 5 cancers-15-05608-t005:** Recommendations for reporting of PGPVs.

Group A					
*ATM*	*BRCA1*	*BRCA2*	*BRIP1*	*CHEK2*	*MLH1*
*MSH2*	*MSH6*	*MUTYH* *	*PALB2*	*PMS2*	*RAD51C*
*RAD51D*	*RET*	*SDHB*	*SDHC*	*SDHD*	*TSC2*
*VHL*					
Group B					
*APC*	*CDH1*	*MEN1*	*NF2*	*PTEN*	*RB1*
*SMAD4*	*STK11*	*TP53*	*TSC1*	*WT1*	

* when biallelic.

## Data Availability

The data can be shared up on request.

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
