# Peer review of "Homologous Recombination Repair Gene Alterations Are Associated with Tumor Mutational Burden and Survival of Immunotherapy"

_cancers, 2023, doi:10.3390/cancers15235608_

Round 1

Reviewer 1 Report (Previous Reviewer 1)

Comments and Suggestions for Authors

The Authors have addressed all my concerns and I have no further comments 

Comments on the Quality of English Language

The Authors have addressed all my concerns and I have no further comments 

Reviewer 2 Report (Previous Reviewer 2)

Comments and Suggestions for Authors

Authors have satisfactorily responded to all questions and criticisms raised.

Reviewer 3 Report (Previous Reviewer 3)

Comments and Suggestions for Authors

the authors partially corrected the work

Comments on the Quality of English Language

the authors partially corrected the work

This manuscript is a resubmission of an earlier submission. The following is a list of the peer review reports and author responses from that submission.

Round 1

Reviewer 1 Report

Comments and Suggestions for Authors

The mansucript entitled "HRR Gene Alterations Predict Tumor Mutational Burden and Sensitivity to Immunotherapy" highlighted that HRR gene alterations can be a potential biomarker predicting TMB-high and a good response to immunotherapy. For successful sequencing, samples with lower percentages of tumor cell nuclei and previous neoadjuvant chemotherapy should be avoided.

-. The Authors should provide the expand forms for all acronyms, including gene acronyms, through the text when they first appear.

- Gene acronyms should be written in italics.

Author Response

(The same comments are written in the attachment)

 - The Authors should provide the expand forms for all acronyms, including gene acronyms, through the text when they first appear.

- Gene acronyms should be written in italics.

Thank you for reviewing and for your kind suggestion.

In addition to spelling out acronyms in the text, we added a section for abbreviations. We corrected gene acronyms in italics, including in figures.

Reviewer 2 Report

Comments and Suggestions for Authors

In this study, Ito and colleagues performed the mutation profile of 204 cancer patients using a NGS-based multigenic panel test. A correlation between alterations in genes involved in homologous recombination repair mechanisms, a high mutational burden, and better overall survival was found. Cases carrying the mutated ARID1A were reported to present a better survival as compared with ones without mutations in this gene.

The methodological plan is appropriate and well conducted; data are well presented.

However, some critical drawbacks make the work weaker:

1) the patients’ collection is too heterogeneous, including tumors with different rates of responsiveness to immunotherapy (altogether, excluding the four most prevalent malignancies, they represent about 15% of the series). Moreover, even making an effort to consider the metastatic cases (disease stage IV) as a group with less heterogeneity – though their anatomical localization and the overall metastasis load may be different for the various cancer types in terms of biological and clinical behavior -, stages I-II and III may have completely different clinical evolution according to the tumor type. The suggestion is to only focus on the most representative carcer types: breast cancer (by stratifying them also for the distinct subtypes according to histology and receptor status) and gastrointestinal + pancreatic carcinomas.

2) extrapolation of the prognostic role for the few ARID1A mutants is too risky due to either the limited number of cases evaluated either to the fact that this gene cooperates with the others ones for the genome repair by homologous recombination mechanisms. Therefore the correlation with the HRR mutants in toto remains the only valid one.

3) it is difficult to understand the utility for readers to include into the manuscript the paragraph “Presumed germline pathogenic variants (PGPVs) and germline testing” when only four cases had genetic counseling.

Minor point. The 18 cases facing failure of the test could be removed by Table 1 (on the contrary, they are confounding)

Reviewer 3 Report

Comments and Suggestions for Authors

the work is potentially interesting

Genes and mutations in various tumors were investigated

1. in the introductory part, it is necessary to add the importance of simultaneous examination of analyzes and multiomics principles of gene mutations that were suggested earlier and their interrelation as suggested in previous works: PMID: 33131355

2. in the introductory part, it is not clear which immune therapy is suggested and the part of the work on immune therapy is not very clearly written. If we mean blockers of key points, then PD-PDl-1 is determined. If you mean targeted therapy, then genes are tested, so you should be very clear. if he means new drugs that induce apoptosis and which are constantly being synthesized and show in vitro effects on apoptosis as shown in papers PMID: 36695998, then it should be clarified.

3. The gene changes that regulate the key points of these molecules have not been investigated, so the introduction should be changed and corrected and harmonized with the gene changes that have been analyzed. There are many gene changes that can be targeted in therapy. Methylations were not mentioned, nor were apoptosis regulators as previously published : PMID: 36378420

4. in table 1, present next to the numbers the percentage of representation, because the numbers mean nothing but the frequency

5. the title of the paper talks about prediction, but nowhere are patients analyzed depending on the clinical stage.

6. It is necessary to present all parameters in relation to the clinical stage of the disease using the TNM classification

7. At the same time, it would be nice to determine the percentage of primary tumors for each disease and the percentage of metastases

8. In the material section, the authors present the finding that about 100 patients were in an advanced stage, and what happened to the others, because the study included twice as many patients. Therefore, divide patients with advanced stages and those who were not in an advanced stage of the disease.

9. in the conclusion, explain the sensitivity and specificity of the methods that were used because many methods examine mutations but with different specificity and sensitivity as shown in the previous work, as well as that the distribution of the methods is not the same everywhere in all parts of the world PMID: 33973139, PMID: 37801920 so interpret the possibilities of meaning in concrete conditions

Comments on the Quality of English Language

increase quality
